# Family Income at Birth and Risk of Attention Deficit Hyperactivity Disorder at Age 15: Racial Differences

**DOI:** 10.3390/children6010010

**Published:** 2019-01-14

**Authors:** Shervin Assari, Cleopatra Howard Caldwell

**Affiliations:** 1Center for Research on Ethnicity, Culture and Health, School of Public Health, University of Michigan, Ann Arbor, MI 48104, USA; cleoc@umich.edu; 2Department of Psychiatry, University of Michigan, Ann Arbor, MI 48104, USA; 3Department of Psychology, University of California, Los Angeles (UCLA), Los Angeles, CA 90095, USA; 4Department of Health Behavior and Health Education, School of Public Health, University of Michigan, Ann Arbor, MI 48104, USA

**Keywords:** class, race, socioeconomic status, socioeconomic position, income, poverty status, ethnicity, Blacks, ethnicity, Attention Deficit Hyperactivity Disorder (ADHD)

## Abstract

Background: Socioeconomic status (SES) resources protect children and adults against the risk of medical and psychiatric conditions. According to the Minorities’ Diminished Returns theory, however, such protective effects are systemically weaker for the members of racial and ethnic minority groups compared to Whites. Aims: Using a national data set with 15 years of follow up, we compared Black and White youth for the effects of family SES at birth on the risk of Attention Deficit Hyperactivity Disorder (ADHD) at age 15. Methods: The Fragile Families and Child Wellbeing Study (FFCWS, 1998–2016) is a longitudinal prospective study of urban youth from birth to age 15. This analysis included 2006 youth who were either White (*n* = 360) or Black (*n* = 1646). The independent variable was family income, the dependent variable was ADHD at age 15. Child gender, maternal age, and family type at birth were covariates, and race was the focal moderator. We ran logistic regressions in the overall sample and specific to race. Results: In the overall sample, high family income at birth was not associated with the risk of ADHD at age 15, independent of all covariates. Despite this relationship, we found a significant interaction between race and family income at birth on subsequent risk of ADHD, indicating a stronger effect for Whites compared to Blacks. In stratified models, we found a marginally significant protective effect of family SES against the risk of ADHD for White youths. For African American youth, on the other hand, family SES was shown to have a marginally significant risk for ADHD. Conclusions: The health gain that follows family income is smaller for Black than White families, which is in line with the Minorities’ Diminished Returns. The solution to health disparities is not simply policies that aim to reduce the racial gap in SES, because various racial health disparities in the United States are not due to differential access to resources but rather the impact of these resources on health outcomes. Public policies, therefore, should go beyond equalizing access to resources and also address the structural racism and discrimination that impact Blacks’ lives. Policies should fight racism and should help Black families to overcome barriers in their lives so they can gain health from their SES and social mobility. As racism is multi-level, multi-level interventions are needed to tackle diminished returns of SES.

## 1. Introduction

Attention Deficit Hyperactivity Disorder (ADHD) is a mental disorder with childhood onset. ADHD can persist throughout the course of life [1]. As ADHD is characterized by impulsivity and inattention, it causes impairment across multiple domains [2]. It increases the risk of several negative outcomes such as poor education and substance abuse [3]. Although the ADHD etiology is complex, both heritable factors [4] and environmental and social influences [5] contribute to its causation. Low socioeconomic status (SES) is one of the major risk factors for ADHD [6]. Financial difficulty, insecure housing, unmarried family type, and low maternal age at birth of child increase the risk of ADHD [7]. Among the environmental and SES risk factors, and similar to some other childhood mental health conditions [8,9], poverty and financial difficulty are among the strongest risk factors for ADHD [7,10,11,12,13].

Although high SES is protective for population health [14,15,16,17,18,19,20,21], the health gain that follows SES is not identical across various social groups [22,23]. Although high family SES reports better health [24,25] and financial strain and poverty increase [26] poor health outcomes, these effects are not similar for both privileged and under-privileged social groups [22,23]. In the United States, racial and ethnic minority groups show smaller health gains from the very same SES resources compared to Whites [22,23]. This is also true for income [27,28], one of the most established SES resources, which has a strong effect on the health of the population and individuals [29,30,31,32,33].

Although high SES tends to reflect better overall health [34,35,36], this protective effect varies across racial and ethnic groups [22,23,37,38,39]. Given that society treats social groups differently, SES has different impacts on purchasing power and access to resources and goods for various racial groups, and ultimately some groups can better translate their available SES resources into tangible outcomes compared to others [40,41,42]. These diminished returns may be due to the fact that upward social mobility has proven to be more difficult for Blacks compared to Whites, due to racism and segregation [22,23,43,44,45], or high levels of stress and discrimination faced by high SES Blacks [40].

Minorities’ Diminished Returns (MDR) of SES [22] and the unequal effect of SES on health across racial groups [23] are among the neglected mechanisms that contribute to the pervasive racial gap in health. Both theoretical [22,23] and empirical [46,47,48,49,50,51,52] work has shown that these diminished returns are robust and can be found regardless of SES resource, health outcome, age group, setting, and cohort [43,44,46,47,48,49,50,51,52]. For example, MDR of SES have been shown on drinking alcohol [46], smoking [39], obesity [53], depression [47], anxiety [54], suicide [48], chronic medical conditions [47], and mortality [49,50,51,52] for Blacks as compared to Whites. An extreme example of these diminished returns is when high SES increases the risk of poor mental health for Black males [27,47,55,56].

High SES is shown to more significantly reduce the risk of medical [53] and psychiatric [47] chronic conditions for Whites than Blacks. For example, the risk of depression may actually be higher, not lower in high SES Black males [27,47,55,56]. Similarly, high income and education reduce the risk of depression [27,47,55,56], obesity [53], and asthma [57] for White but not Black youth and children. Although asthma, obesity, and depression are all comorbidities of ADHD, we are not aware of any similar studies on the differential effects of SES indicators on the risk of ADHD.

### Aims

To enhance the current knowledge regarding the applicability of MDR theory [22,23] to racial disparities in ADHD in youth, we compared Blacks and Whites who were under follow up from their birth to age 15 for the protective effect of family income at birth on the risk of ADHD at age 15. Although we know that both race [58,59,60] and SES [6,7] impact the incidence and prevalence of ADHD, we are not aware of any previous longitudinal studies that have tested the multiplicative effects of race and SES on the risk of ADHD. So, it is still unclear whether it is “race or poverty” or “race and poverty” that shapes the ADHD disparities in the US. In line with theoretical [22] and empirical [57] work, we expect smaller effects of family income in terms of preventing ADHD for Black compared to White children.

## 2. Methods

### 2.1. Study Design

The Fragile Families and Child Wellbeing Study (FFCWS; 1998–2016) is a 15-year longitudinal study that enrolled a random sample of economically fragile families from 20 US cities with a population of at least 200,000 individuals. Although a detailed description of the study design, conceptual model, sampling, methodology, and data collection of the FFCWS is available elsewhere [61,62,63,64], we provide a brief summary of the FFCWS methodology here.

### 2.2. Study Sample

The FFCW original sample included 4655 families, composed of 2407 Black, 1354 Hispanic, and 894 White families. The FFCWS oversampled non-married couples. As a result, the findings of the FFCWS are not representative of the US general population as a whole. The results are relevant to minority, non-marital, and low SES families.

### 2.3. Ethical Considerations

The Princeton University (PU) Institutional Review Board (IRB 13-1946) approved the FFCWS study protocol. All the adolescents’ legal guardians, parents, and primary caregivers signed informed consent. All youth provided assent. All the participants received financial incentives to compensate for their time.

### 2.4. Current Analysis and Analytical Sample

The current analysis used data from the first and sixth waves of the study. We only included Black and White families who had data on baseline family income and youth chronic disease (ADHD) 15 years later. The analytical sample in this paper was 2006 youth who were either White (*n* = 360) or Black (*n* = 1646).

### 2.5. Study Variables

Independent Variable: Family income at birth was the main independent variable. This variable was calculated based on the family income reported by the mother. Family income was operationalized as an interval measure, with high scores indicating high SES.

Dependent Variable. Primary caregivers of the child/youth were asked: “Has a doctor or health professional ever told you that YOUTH has ADHD?” Responses included yes, no, refuse to answer, do not know.

Covariates: Child gender, family structure, and maternal age at the time of birth (wave 1) were measured as confounders. Child gender was a dichotomous variable (male 1, female 0). Maternal age was an interval variable, treated as a continuous measure. Family structure was a dichotomous variable that was calculated based on the mothers’ report (0 not married, 1 married).

Moderator: Race/ethnicity was the focal moderator/effect modifier in the current study. Race was operationalized as a dichotomous variable, with Whites as the reference category (score 0) and Blacks coded as 1.

### 2.6. Data Analysis

For our data analyses, we used Stata version 15.0 (Stata Corp; College Station, TX, USA). For univariate analysis, we reported mean and proportions in the pooled sample as well as by race/ethnicity. Taylor series linearization was applied to re-estimate the design-based standard errors (SEs). As a result, all the inferences reported here consider sampling weights. We used sub-pop survey commands. We ran our analyses in the overall sample and for each race/ethnic group. For bivariate analysis, we applied the Pearson Chi-square test as well as an independent sample *t* test to compare Whites and African Americans at baseline and at age 15. For multivariable models, we used four logistic regressions. From our logistic regressions, we reported adjusted odds ratios (ORs), standard errors (SE), 95% confidence intervals (CIs), and *p* levels. Four logistic regression models were fitted to the data. In all of our models, ADHD at age 15 was the main dependent variable, family income was the main independent variable, while child gender, maternal age, and family type (i.e., marital status of the family at birth) were covariates. Model 1 and Model 2 were estimated in the overall sample, while the last two models were conducted for each racial/ethnic group. Model 1 did not include the interaction term; however, the second model also included a race by poverty status interaction term.

## 3. Results

### 3.1. Study Participants

This study included 2006 youth who were under follow up from their birth to age 15. Participants were either White (*n* = 360) or Black (*n* = 1646).

Table 1 provides a description of the pooled sample, as well as for each race/ethnic group. Black children had lower maternal age at birth compared to White children. Although the majority of the White children were born to families in which parents were married, most Black children were born to unmarried families. Family income was substantially lower for Black youth compared to White youth. Finally, the prevalence of ADHD at age 15 was not significantly different between White and Black youth (Table 1).

### 3.2. Regression Models in the Overall Sample

Table 2 summarizes the results of Model 1 and Model 2 that were performed in the overall sample. The first model did not include any interaction term and did not show any association between family income at birth and risk of ADHD at age 15. The next model showed a significant interaction between race and family income on the risk of ADHD, indicating a significantly smaller protective effect for Blacks than Whites.

### 3.3. Regression Models Specific to Each Race

Table 3 summarizes the results of Model 3 and Model 4 in racial groups. Our stratified regression models revealed a marginally significant protective effect of family income against the risk of ADHD for White, but a marginally significant risk effect for Black youth (Table 3).

## 4. Discussion

The current study showed a differential effect of family income at birth on the risk of ADHD at age 15 for White and Black youth. High family income at birth reduced ADHD risk for White but not Black youth at age 15.

In the National Survey of Children’s Health (NSCH, 2003–2004), which included 86,537 Black and White children (17 years old or younger), a higher SES was associated with a lower risk of childhood asthma for both Whites and Blacks. The magnitude of this protective effect was considerably larger for White compared to Black children and youth [57]. Similar patterns are shown for depression [27,47,55,56], obesity [53], and asthma [57], meaning that SES better protects Whites than Blacks for several mental health and health outcomes.

The protective effect of high family income against ADHD for White youth is in line with previous studies [6,7], showing that there is an economic gradient in the epidemiology of ADHD and other health outcomes. The same pattern is shown for other chronic conditions [65]. This protection is, however, missing for Blacks. There are multiple studies showing that high SES of the family generates fewer health protections for Black than White children. A series of analyses on the Fragile Families and Child Wellbeing Study (FFCWS) data documented Black–White differences in gaining health from family SES. The effects of family SES at birth on future BMI [53], self-rated health [66], impulsivity [67], and chronic disease [57] are smaller for Blacks than Whites, which is in line with the MDR theory [53].

The finding that equal family income at birth generates unequal protection for White and Black youth is similar to the findings of a large number of studies that have documented diminished effects of family SES on obesity, impulse control, subjective health, and asthma [53,57,66,67]. This is in part because educated Blacks are more likely to remain poor compared to educated Whites [68,69].

Although MDR theory is well supported by empirical data [22,23], fewer studies have focused on children and youth [70,71] as compared to adults [41,47,48,72]. Among the various possible underlying mechanisms that may cause smaller health gains of SES among Blacks than Whites, childhood characteristics and differential exposure early in life may play some role [53]. Smaller health gains from family SES at birth may be one of the reasons Black children suffer worse health outcomes than White children [67,68]. This mechanism is different from Black–White mean difference in exposures to risk factors and access to resources and buffers such as the health care system [73,74]. In this view, socially, economically, and politically disadvantaged minority groups unequally gain from their very same resources compared to the economically, politically, and socially privileged majority group.

Diminished health effects of SES for Blacks do not suggest that Black families are less able to effectively take advantage of their human and economic resources and turn them into tangible outcomes. Such an argument would be blaming the victim. Reduced gains are not because of their culture, but because they have been marginalized, stigmatized, and treated poorly in this society. Not only do Black individuals have a lower chance of upward social mobility, they do not enjoy similar benefits from their social mobility [43,44,45]. Such diminished returns are a function of racism and discrimination [75], rather than the culture of poverty [76]. These diminished returns may be a result of being victimized and stigmatized over generations. Some of the minorities may feel some pressure to appear strong, wealthy, and successful [77,78,79,80,81,82,83,84,85,86,87], which may have an impact on their children. Instead of individual behaviors, choices, and preferences contributing to success, it is the US social structure that can cause MDR [22,23]. Due to segregation and institutional and structural racism, Black individuals face far more societal barriers in their daily lives that hinder them from obtaining the most possible health gains from each available resource across generations.

It is not Black families, but the current US political and social system that has failed, resulting in differential patterns of health and mental health outcomes [22,23]. US society consistently demands more of high SES Blacks so that they pay an extra psychological and physiological cost for their success. In race-aware societies that treat people differently based on their skin color, upward social mobility is not similarly feasible for all racial groups. Blacks pay far more social, psychological, and physiological costs for the same social success compared to Whites [43,44]. The current political, economic, and social systems in the US are designed to maximize the gains of privileged groups. This, of course, comes with costs to non-White groups [22,23]. This is one of the many reasons why closing the SES gap does not close the stubborn Black–White health gap.

We used family income at birth instead of income at age 15 for several reasons. First, we were particularly interested in causal effects, which requires temporal order (exposure occurs before outcome). We preferred to test the consequence of exposure 15 years later. In addition, FFCWS is one of the few long-term longitudinal steadies and we wanted to take advantage of the longitudinal design of the study. Furthermore, most of the literature is on cross-sectional associations rather than on longitudinal effects. So, with this decision, our paper made a unique contribution to the field. Finally, the diminished returns of income and other SES require time to develop. That is, over the 15 years from birth, the groups develop differences in the effects of SES, which may not be present when cross-sectional associations are tested. This is in line with a cumulative disadvantage approach, which is also in line with the life course developmental approach. Finally, we were interested in the intergenerational transmission of diminished returns of SES. As a result, we used family income at birth instead of the SES of the participant at age 15.

### Limitations

The study had a few limitations. First, similar to any other long-term cohort, the study had a loss to follow up and attrition, which was not at random. The outcome was not confirmed based on administrative data, but the self-report of parents and caregivers. In addition, the FFCWS does not have any variable that reflects the possibility of heredity of ADHD. We have added this as a missing covariate in this study’s limitations section. Finally, the results of our analysis should be interpreted with caution because: 1) there is only a limited number of controls; and 2) the focus is exclusively on Black–White gaps in ADHD, while other types of categorical comparisons are not the subject of this study (for example, the FFCWS contains a relatively large sample of Hispanics, but the authors did not include this group in their analysis). Therefore, the generalization of findings may be limited. Despite these limitations, the results make a unique contribution as very few studies, if any, have focused on the diminished returns of SES indicators on ADHD.

## 5. Conclusions

The protective effect of family income at birth in lowering the risk of ADHD at age 15 differs for White and Black youth. This is a longitudinal replication of a cross-sectional finding [57], suggesting that there are more high SES Blacks with ADHD than high SES Whites with the same condition. Diminished returns of SES is a systemically neglected contributor to racial gaps in health in the US. Future research should decompose the effects of structural racism, interpersonal discrimination, stress, environmental exposures, neighborhood SES, and societal barriers that may have some role in these heterogenetic associations. At the same time, public, social, and economic policies should go beyond equalizing SES and income and instead equalize the impact of such resources. Policies should explore multi-level solutions that eliminate or at least reduce the MDR of SES resources. A realistic solution to the racial gap in health requires the joint consideration of race and SES, which means not ignoring the systemic interactions between race and SES and their role in shaping health disparities.

## Figures and Tables

**Table 1 children-06-00010-t001:** Sociodemographic characteristics in the pooled sample and across race/ethnic groups.

	All (*n* = 2006)		Whites (*n* = 360)		Blacks (*n* = 1646)	
	**% (*SE*)**	**95% *CI***	**% (*SE*)**	**95% *CI***	**% (*SE*)**	**95% *CI***
Child Gender ^a^						
Female	44.77 (0.03)	37.70–51.84	47.35 (0.06)	35.27–59.42	40.96 (0.05)	31.62–50.30
Male	55.23 (0.03)	48.16–62.30	52.65 (0.06)	40.58–64.73	59.04 (0.05)	49.70–68.38
Family Type at Baseline *^a^						
Not-Married	36.62 (0.02)	31.62–41.62	15.93 (0.03)	10.67–21.18	67.24 (0.05)	57.39–77.09
Married	63.38 (0.02)	58.38–68.38	84.07 (0.03)	78.82–89.33	32.76 (0.05)	22.91–42.61
ADHD at Age 15 ^a^						
No	80.19 (0.04)	72.15–88.24	80.60 (0.05)	71.41–89.78	79.59 (0.06)	67.00–92.18
Yes	19.81 (0.04)	11.76–27.85	19.40 (0.05)	10.22–28.59	20.41 (0.06)	7.82–33.00
	***Mean* (*SE*)**	**95% *CI***	***Mean* (*SE*)**	**95% *CI***	***Mean (SE)***	**95% *CI***
Maternal Age at Baseline *^b^	27.46 (0.47)	26.50–28.42	29.42 (0.52)	28.36–30.48	24.55 (0.85)	22.82–26.28
Family Income at Baseline *^b^	49.35 (2.65)	43.94–54.75	64.51 (4.30)	55.75–73.28	26.90 (2.20)	22.42–31.38

Notes: Source: Fragile Families and Child Wellbeing Study (FFCWS; 1998–2016), * *p* < 0.05 for Blacks versus Whites comparisons, ^a^ Chi Square Test, ^b^ Independent Samples *t* Test, Attention Deficit Hyperactivity Disorder (ADHD), Standard Error (SE), Confidence Interval (CI).

**Table 2 children-06-00010-t002:** The results of logistic regressions in the overall sample.

	OR (*SE*)	95% *CI*	*T*	*p*
**Model 1 (All) (*n* = 2006)**				
Race (Black)	0.84 (0.52)	0.24–2.98	−0.27	0.787
Child Gender (Male)	2.43 (1.15)	0.93–6.37	1.88	0.069
Maternal Age at Baseline	0.97 (0.05)	0.88–1.07	−0.65	0.519
Family Type (Married) at Baseline	1.60 (1.34)	0.29–8.78	0.56	0.580
Family Income at Baseline	0.99 (0.01)	0.98–1.01	−0.95	0.349
Intercept	0.37 (0.48)	0.03–5.13	−0.77	0.446
**Model 2 (All) (*n* = 2006)**				
Race (Black)	0.22 (0.14)	0.06–0.79	−2.41	0.022
Child Gender (Male)	2.52 (1.23)	0.93–6.79	1.90	0.067
Maternal Age at Baseline	0.97 (0.05)	0.87–1.08	−0.60	0.550
Family Type (Married) at Baseline	1.43 (1.11)	0.30–6.94	0.46	0.646
Family Income at Baseline	0.98 (0.01)	0.96–1.00	−1.86	0.072
Family Income at Baseline × Race	1.03 (0.01)	1.01–1.06	2.99	0.005
Intercept	0.71 (0.97)	0.04–11.47	−0.25	0.805

Notes: Source: Fragile Families and Child Wellbeing Study (FFCWS; 1998–2016), Outcome: Attention Deficit Hyperactivity Disorder (ADHD) at age 15, Confidence Interval (CI).

**Table 3 children-06-00010-t003:** The results of two logistic regression models by race.

	OR (*SE*)	95% *CI*	*t*	*p*
**Model 3 (Whites) (*n* = 360)**				
Child Gender (Male)	2.07 (1.39)	0.53–8.11	1.08	0.287
Maternal Age at Baseline	0.97 (0.06)	0.85–1.10	−0.50	0.619
Family Type (Married) at Baseline	1.26 (1.18)	0.19–8.49	0.25	0.806
Family Income at Baseline	0.98 (0.01)	0.96–1.00	−1.78	0.084
Intercept	0.88 (1.50)	0.03–28.00	−0.07	0.943
**Model 4 (Blacks) (*n* = 1646)**				
Child Gender (Male)	3.63 (1.52)	1.55–8.51	3.08	0.004
Maternal Age at Baseline	0.98 (0.05)	0.87–1.09	−0.41	0.685
Family Type (Married) at Baseline	1.48 (1.32)	0.24–9.15	0.44	0.665
Family Income at Baseline	1.02 (0.01)	1.00–1.03	2.38	0.024
Intercept	0.10 (0.12)	0.01–1.08	−1.97	0.058

Notes: Source: Fragile Families and Child Wellbeing Study (FFCWS; 1998–2016), Outcome: Attention Deficit Hyperactivity Disorder (ADHD) at age 15, Confidence Interval (CI), Odds Ratio (OR).

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
