# Peer review of "Family Income at Birth and Risk of Attention Deficit Hyperactivity Disorder at Age 15: Racial Differences"

_children, 2019, doi:10.3390/children6010010_

Round 1
Reviewer 1 Report
This paper is very interesting and deals with an important, though neglected issue of SES and its influence according to ethnic basis However, I have some comments:.
Editorial, for example in line 46 ADHD, not ADHE and in line 82 - previous, not precious
It should be noted that asthma, obesity and depression are all comorbidities of ADHD, which makes these examples even more intriguing concerning the question of ADHD
Methods:The possibility of heredity, which is quite common in ADHD should, I think, be included in the covariates since it could have an effect on the results (that would be interesting to study). Other interesting convariates could be obesity asthma and depression, all mentioned in the introduction according to the same study question
Discussion: The discussion is interesting, but a bit narrow-spanned, which is regrettable. These results are interesting enough to open the discussion a bit more, including the mentioned above: the possibility of heredity? a relationship between obesity depression and perhaps asthma and ADHD, which is already discussed in the literature?. Also, completely not in line of blaming the victims, which means gaining less because of their culture, there is also another explanation that is exhibited in other minority populations as well. Sometimes, these minorities exhibit a need to show themselves as stronger, better and healthier, which might result in more pressure on the children. It is also in line with the studies that show less tendency to be treated in different minority groups. Far from stigmatization, it could also be a cost of being victimized and stigmatized for so long, and I believe that it should be mentioned as well.
Author Response
Thank you for your very positive evaluation and constructive comments. Here are the changes and responses. All the changes are in yellow color.
Background:
1- ADHE --> ADHD
2- Precious --> previous
3- We are now mentioning in the introduction that asthma, obesity and depression are all comorbidities of ADHD, so the literature on these examples are very relevant to ADHD.
Methods:
1- We have added this as a missing covariate in the study limitation section:
2- The FFCWS does not have any variable that reflects the possibility of heredity of the ADHD.
3- For various reasons (explained below), we did not include obesity, asthma, and depression. First the mechanism for diminished returns are similar, so controlling for them would bias the results toward the null, and will be a case of over- or unnecessary adjustment. Second, the paper becomes similar to those we have published on asthma, depression, and obesity. So, some of the statistics become redundant. Finally, all our covariates are at baseline. There are no variables on obesity asthma and depression at baseline (to be controlled for). That will change our conceptual model.
Discussion:
1- We added three sections to the paper:
2- We added sentences on the heredity as well as comorbidities (obesity, depression and asthma) to our discussion section.
3- We mentioned that these diminished returns may be a result of being victimized and stigmatized for so long time.
4- Some of the minorities may feel a need to show themselves as strong and healthy which might result in more pressure on their children.
Reviewer 2 Report
Using a sample of youth who were either non-Hispanic white (n = 360) or black (n = 1646) from the from the Fragile Families and Child Wellbeing Study (FFCWS), 1998 - 2016, a 15-year longitudinal study of urban children and youth from birth to age 15, this article explores black-white gap in the risk of ADHD at age 15 as a function of family income, maternal age, child gender, and family type at birth. The author found that, net of controls (maternal age, child gender, and family type at birth), there is a weak association between household income at birth and the risk of ADHD at age 15, and this association differs for whites and blacks. In other words, black children from higher-income families are more likely to suffer from ADHD than white children of the same income. The author ties the results to the Minorities’ Diminished Return theory, which contends that the protective effect of SES on health is weaker for the racial and ethnic minorities than for majority whites. However, these types of analysis should be interpreted with caution because: (1) there is only a limited number of controls; and (2) the focus is exclusively on white-black gap in ADHD, while other types of categorical comparisons are not subject of this study (for example, the FFCWS contains a relatively large sample of Hispanics, but the author did not include this group in their analysis). Therefore, generalization of findings may be limited. It is also worth mentioning that the order of citations is wrong throughout the manuscript.
Author Response
Thank you for your comments. They have helped us have a better paper, particularly on our limitation section.
We added a few sentences on the issue of generalizability.
The results of our analysis should be interpreted with caution because: (1) there is only a limited number of controls; and (2) the focus is exclusively on white-black gap in ADHD, while other types of categorical comparisons are not subject of this study (for example, the FFCWS contains a relatively large sample of Hispanics, but the author did not include this group in their analysis). Therefore, generalization of findings may be limited.
The reference order will be fixed with the MDPI staff after acceptance. I am always thankful of them for this service, which saves authors time and resources.
Reviewer 3 Report
This study examines the association between
In the introduction part, the author reviewed previous literature on family SES and health. However, since this study is solely focused on family household income, and did not include any education or occupation indicators, it is vital that the author provide a more focus review of previous findings on the link between family household income and health. Literature has suggested differential effects of family income, parental education, and other indicators on childhood/adult health outcomes.
The author mentioned the lack of previous studies on SES and ADHD among children/youth, which seemed rather surprising to the reviewer. But a quick search on Google Scholar, I was able to find a few articles on similar topic, including a systematic review (https://link-springer-com.libproxy.temple.edu/article/10.1007/s10578-015-0578-3). Previous findings would be essential in helping the author formulating hypothesis, and in helping readers understand the context of this study. This is a major issue that need to be addressed.
Is any weighting applied to the analysis? This needs to be addressed.
Why was income at birth used, instead of income at age 15?
Maternal education was mentioned on page 3, but never included in analysis. Any reasons why? More importantly, studies have consistently suggested that paternal educational attainment is a strong indicator of childhood health outcomes. Any theoretical or methodological reasons why paternal education was not included?
In table 2, as is in multiple places throughout the manuscript, the author used "SES (family income)." I wonder why the author would not simply use family income. Family SES usually usually suggests multiple indicators. Simply referring to it as it is, i.e., family income, would be less misleading.
Minor issues:
The semi-colon in the title seemed to have been misplaced. Should it be a colon?
The references were not presented in order. For example, it started with No. 111 on page 1. This would need to be formatted better.
Author Response
Again, please be aware this paper is built on Minorities’ Diminished Returns (MDR) theory, which is on SES on income. So, income is just 1/10th of the literature on MDR. Still, to accommodate this important comment, I added some of the literature on the health effects of income. These are the new citations.
· Marmot M. The influence of income on health: views of an epidemiologist. Health Aff (Millwood). 2002 Mar-Apr;21(2):31-46.
· Deaton, A. Health, income, and inequality. National Bureau of Economic Research Reporter: Research Summary.. 2003
· Lynch J, Smith GD, Harper S, Hillemeier M, Ross N, Kaplan GA, Wolfson M. Is income inequality a determinant of population health? Part 1. A systematic review. Milbank Q. 2004;82(1):5-99.
· Ecob R, Smith GD. Income and health: what is the nature of the relationship? Soc Sci Med. 1999 Mar;48(5):693-705.
· Lynch J, Smith GD. Commentary: Income inequality and health: the end of the story? Int J Epidemiol. 2002 Jun;31(3):549-51.
We added a more complete literature review to our paper.
These are the new papers added:
· Russell G, Ford T, Rosenberg R, Kelly S. The association of attention deficit hyperactivity disorder with socioeconomic disadvantage: alternative explanations and evidence. J Child Psychol Psychiatry. 2014 May;55(5):436-45. doi:10.1111/jcpp.12170.
· Russell G, Ford T, Rosenberg R, Kelly S. The association of attention deficit hyperactivity disorder with socioeconomic disadvantage: alternative explanations and evidence. Journal of Child Psychology and Psychiatry, 2013.
· Lingineni RK, Biswa S, Ahmad N, Jackson BE, Bae S, Singh KP. Factors associated with attention deficit/hyperactivity disorder among US children: Results from a national survey. BMC Pediatrics, 2012. 12(1): p. 50.
· Russell AE, Ford T, Williams R, Russell G. The Association Between Socioeconomic Disadvantage and Attention Deficit/Hyperactivity Disorder (ADHD): A Systematic Review. Child Psychiatry Hum Dev. 2016 Jun;47(3):440-58. doi: 10.1007/s10578-015-0578-3.
· Russell AE, Ford T, Russell G. Socioeconomic Associations with ADHD: Findings from a Mediation Analysis. PLoS One. 2015 Jun 1;10(6):e0128248. doi: 10.1371/journal.pone.0128248.
· Petrill SA, Pike A, Price T, Plomin R. Chaos in the home and socioeconomic status are associated with cognitive development in early childhood: Environmental mediators identified in a genetic design. Intelligence, 2004. 32(5): p. 445–460.
· Reiss F. Socioeconomic inequalities and mental health problems in children and adolescents: A systematic review. Social Science & Medicine, 2013.
We added to our literature review on SES – ADHD.
We mentioned in the statistical section that the weights are applied.
We explained in an additional paragraph why was income at birth used instead of income at age 15? This is the paragraph:
We used income at birth instead of income at age 15 for several reasons. First, we were particularly interested in causal effects, which requires temporal order (exposure occurs before outcome). So we preferred to test the consequence of exposure 15 years later. In addition, FFCWS is one of the few long term longitudinal steadies and we wanted to take advantage of the longitudinal design of the study. Furthermore, most of the literature is on cross-sectional associations rather than longitudinal effects. So, with this decision, our paper made a unique contribution to the field. Finally, the diminished return of income and other SES require time to develop. That is, over the 15 years from birth, the groups develop differences in the effects of SES, which may not be present when cross-sectional associations are tested. This is in line with a cumulative disadvantage, which is also in line with the life course developmental approach. Finally, we were interested in the intergenerational transmission of diminished returns of SES. As a result, we used SES of parents at birth instead of the SES of the participant at age 15.
Maternal education on page 3 was a typo. We did not want to include maternal education in this paper. This is become income explains some of the effects of education, so controlling for education will be unnecessary or over adjustment (Schisterman EF, Cole SR, Platt RW. Overadjustment bias and unnecessary adjustment in epidemiologic studies. Epidemiology. 2009 Jul;20(4):488-95. doi: 10.1097/EDE.0b013e3181a819a1.).
We used the combination SES (income) because diminished return theory Is about SES not just income. However, this is a good point, and to make the paper more accurate, we are now using family income rather than SES (income).
Minor issues:
The semi-colon in the title is now replaced with a colon.
Thanks to the MDPI system, one of the very significant advantages of publishing with them is that they order the references after acceptance. I really appreciate this service as it saves considerable time for me. So, all of the references will be in order at the time of publication.
Round 2
Reviewer 3 Report
The responses adequately addressed the comments. I have no further comments.